# Does COVID-19 Affect the Behavior of Buying Fresh Food? Evidence from Wuhan, China

**DOI:** 10.3390/ijerph18094469

**Published:** 2021-04-22

**Authors:** Jing Chen, Yong Zhang, Shiyao Zhu, Lei Liu

**Affiliations:** 1School of Transportation, Southeast University, No. 2, Southeast University Road, Jiangning District, Nanjing 211189, China; zhangyong@seu.edu.cn (Y.Z.); lei1992@seu.edu.cn (L.L.); 2School of Civil Engineering, Southeast University, No. 2, Southeast University Road, Jiangning District, Nanjing 211189, China; crystal.zhusy@gmail.com

**Keywords:** COVID-19, fresh food, shopping behavior, Wuhan

## Abstract

COVID-19 first appeared in Wuhan city of Hubei Province in China in December 2019. It has a substantial impact on human life all around the world, especially for citizens. The threat of COVID-19 has resulted in people shopping online to get fresh food and reduce outdoor trips. Collecting data from adult internet users in Wuhan, China in 2020, this study aims to explore the influence of COVID-19 on fresh food shopping behavior. In addition, a comparison and ordered logit model are constructed to demonstrate the changes and effects of COVID-19. The results suggest that more citizens in Wuhan city will buy fresh food online and the cost and frequency are also increased. The experience of online shopping for fresh food during the lock-down days has promoted more online shopping. The factors, such as frequency of online shopping before the COVID-19 outbreak, frequency of online shopping during the COVID-19 pandemic, and age, have a negative effect on the proportion of online shopping after the lock-down days, while the proportion of online shopping before the COVID-19 outbreak, the proportion of online shopping during the COVID-19 pandemic, and travel time of in-store shopping before the COVID-19 outbreak have a positive effect. The results provide insights for managers, city planners, and policymakers.

## 1. Introduction

In December 2019, a cluster of pneumonia cases infected with the novel coronavirus was reported in Wuhan, China. Now, it has rapidly spread across the continents. More than 200 countries/regions have reported confirmed COVID-19 cases, and it turned out to be a major public health issue [1]. Even though we have vaccines now, still the numbers of infections and deaths are rising. Food systems and security face new challenges during the COVID-19 pandemic [2,3,4,5].

Even though COVID-19 can be transmitted from food [6], online shopping can reduce the possibility of being infected by COVID-19 without traveling. Shopping behavior during the COVID-19 pandemic has changed [7] and generally depends on fear [8]. This induced people to shop more online than before and the food reserve time extended from 3.37 to 7.37 days after the COVID-19 breakout [9]. The frequency of in-store shopping for food was low during the COVID-19 pandemic [10]. Leone et al. [11] show the challenges and propose future directions on the research of healthy food retail during the COVID-19 pandemic. They also emphasize that the effects of the COVID-19 pandemic on consumers’ shopping behavior are still unknown. This study aims to enrich the research in this field.

Researchers show interest in the effect of the COVID-19 pandemic on fresh food. However, there is still limited research that investigates the effects of COVID-19 on fresh food shopping behavior. The Canadian fruit and vegetable markets are affected by COVID-19, and these changes may have a long-term effect on fresh fruit and vegetable distribution [12]. Mitchell, Maull, Pearson, Brewer, and Collison address that it is necessary to explore the impact of COVID-19 on the UK fresh food supply chain [13]. The socio-demographic factors that affect buying fresh vegetables have been addressed, and propose that future studies should focus on the effects of before and after the end of the COVID-19 pandemic [14].

Wuhan city was the first place outbreak of COVID-19, the virus spread quickly, and we knew little about it at the beginning. The government decided to lock down the city to prevent the fast spread of COVID-19, from 23 January to 8 April 2020, which lasted 76 days. More than 10 million citizens in Wuhan city had experienced a long and hard fight against the virus. People work from home, buy food online, cannot travel around during the lock-down days. During the COVID-19 pandemic, lots of fresh food from other areas was donated to Wuhan city to help them overcome this hardship. Therefore, in Wuhan, the first city stricken by COVID-19, citizens got fresh food from online shopping and donations for daily life. These experiences have had an impact on their future lives after the city re-opened, especially in buying fresh food. This paper explores the effects of COVID-19 on buying fresh food based on the data collected in Wuhan city.

Hence, there is still a need for more research to investigate the effects of COVID-19 on buying fresh food online. The aim of the present study was to observe whether the COVID-19 pandemic has influenced people’s shopping behavior of fresh food in Wuhan city. The influence factors have been analyzed as well. This paper will enrich the study of the effect of COVID-19 on buying fresh food, and the results will be useful for the cold chain logistics industry.

## 2. Literature Review

### 2.1. Fresh Food

Fresh food is important for people’s daily lives, such as vegetables, fruit, meat, fish, milk, etc., which are consumed every day. Therefore, the shopping frequency for fresh food is much higher than other goods. With the improvement of the living standard, people have higher requirements for the quality of fresh food. Fresh food is not like any other food, the shelf life is shorter than other food. It is easily spoiled during delivery, research on its access [15,16], security [17], and route planning [18,19,20] for fresh food has attracted more attention. People usually get fresh food from the grocery store, but farmers or peddlers will sell fresh food at the door of the community, mainly vegetables and fruits. This is a particular way of getting fresh food in China. However, with the COVID-19 breakout in Wuhan city, no one provided such convenient service for fresh food. Citizens turned to online shopping for fresh food.

There is a great improvement of cold chain logistics in China, which contribute to more often shopping online for fresh food and an increasing demand for fresh food [21]. There are many fresh shopping apps in China, like Freshippo, Missfresh, and eleme, which can deliver fresh food within several hours after customers have paid for the order. Fresh food is easy contaminated during shipping, and the delivery time is closely related to cost and customer satisfaction. Consequently, planning the shortest route for the delivery of fresh food will minimize cost and carbon emissions [20]. With the increase of fresh food sales online, awareness of people’s demand for fresh food can help fresh food E-commerce companies reduce the loss of fresh food that has expired.

### 2.2. Online Shopping and In-Store Shopping

With the development of online shopping and the improvement of Internet technology, more and more people will tend to shop online, which would cause the decline of in-store shopping. In the past 10 years, online shopping has grown immensely in China. In 2018, online retail sales were over 9 trillion RMB (≈US $1.27 trillion) in China, with a growth of 23.9% compared to the previous year. China’s online retail market is still the largest in the world [22]. Online shopping is growing rapidly, its effects on in-store shopping have been studied by many researchers [23,24,25], as well as the relationship [25,26,27,28].

There is considerable empirical work investigating the relationship between online shopping and in-store shopping, and four types (substitution, complementarity, neutrality, modification) of effects are concluded [24,29,30]. As shown in Table 1, both online shopping and in-store shopping have irreplaceable advantages. Entertainment and social interaction play an important role in customers’ in-store shopping, which enhances customers’ store loyalty as well [31,32]. Customers can feel and touch items and communicate with salesclerks, those who like shopping in-stores enjoy leisure time and social interaction there, while online shopping could not provide such experiences. Online shopping may reduce personal trips [24] with making goods accessible by door-to-door deliveries, and has positive effects on the frequencies of shopping [30]. There are no trips to stores with online shopping, travel time and cost saved [24]. Online shopping could be a saving strategy and a leisure-oriented way. What is more, unlike in-store shopping, online shopping has no time limitation, people can shop online at any time. Some other positive factors may result in more people shopping online, such as a faster Internet connection [33]. People can experience and compare the physical goods by in-store shopping, so there are fewer returns on purchases and regrets. The more shopping opportunities that one can reach within 10 min, the lower the frequency of online shopping [29]. On one hand, online shopping may substitute in-store shopping, but on the other hand, in-store shopping may complement online shopping.

Various methods are used to analyze online shopping and in-store shopping behavior, the most widely adopted approaches are structural equation modeling (SEM) [23], binary logit model [35], multinomial regression model [36], etc. Based on the survey data (most of them are questionnaires), positive or negative results will be obtained. In this study, an ordered logit model will be used to explain how online shopping proportion after the COVID-19 pandemic is influenced. The variables mainly contain socio-demographic features, shopping behaviors, Internet behaviors, shopping attitudes, Land use, as shown in Table 2.

Will COVID-19 change the behavior of buying fresh food? Will COVID-19 promote more often buying fresh food online? Is it possible for COVID-19 to change the way of shopping? This study aims to explore the effects of the COVID-19 pandemic on fresh food shopping behavior and try to answer the above questions. First, we describe the difference in fresh food shopping behavior before, during, and after the COVID-19 pandemic. This includes online and in-store shopping frequencies, costs and online shopping proportion, etc. Second, we explain how online shopping proportion after the COVID-19 pandemic is influenced by shopping behaviors before and during COVID-19 pandemic.

## 3. Methodology

### 3.1. Survey Design and Data

The research area contains 13 districts, which are in the capital city of Hubei Province, the largest city in central China [46] (see Figure 1). Wuhan city has an administrative area of approximately 8500 sq.km and a total population of approximately 12 million (2019). It was the first city stricken by the COVID-19 pandemic, the terrible disease infected 50,340 people and took 3869 people’s lives from Wuhan city [47]. Wuhan city was the first one locked down due to COVID-19 pandemic in the world, and the last one reopened in China. Citizens in the city experienced 76 locked-down days. Therefore, we choose this city to explore the effects of the COVID-19 pandemic on buying fresh food.

In this paper, questionnaires were collected from citizens who have experience shopping for fresh food, with answers about essential information of themselves [36] and their behavior characteristics of shopping for fresh food both online and in-store [29]. The variables first selected were from previous studies, as shown in Table 2. Questionnaire were designed based on the variables. The questionnaire survey can be conducted via the field, an interview, mailing it in, or online [7,14,48,49,50,51]. As Wuhan is still threatened by COVID-19, citizens should keep their distance from each other, therefore the survey could not be conducted through face-to-face interviews. We conducted the survey online by Wenjuanxing (a platform providing functions equivalent to Amazon Mechanical Turk), which may conduct the survey through social media, such as QQ and WeChat. The data were collected from the online survey conducted in Wuhan from May 2 to 12 May 2020. There were 171 respondents that participated in this survey. A total of 156 effective questionnaires were collected, which accounts for 91.12%. Procedures of survey design and data collection are draw in Figure 2.

The questionnaire mainly contains four parts. The first one is about socio-demographic characteristics, mainly including gender, age, education, occupation, and income, which are the essential information of themselves. The second is about the information of different shopping choices before the COVID-19 outbreak, mainly including preference for online shopping or in-store shopping, online shopping rate and cost, travel mode and time for in-store shopping, and online and in-store shopping frequency. The third is about online shopping information during the COVID-19 pandemic, mainly including online shopping cost, frequency, rate, time, and the waiting time for purchased fresh food. The fourth is about attitudes of online shopping for fresh food after the COVID-19 pandemic, mainly including the estimate of online shopping proportion for fresh food, advantages and disadvantages of online shopping for fresh food compared with in-store shopping. Detailed survey information is shown in Appendix A
Table A1.

#### Respondents’ Attributes

The basic attributes of respondents in this survey are shown in Appendix A
Table A1. The female respondents account for 53.8%, and male account for 46.2%. Women are found more likely to shop online than men [30,38], especially for daily goods [41]. The distribution of age, however, tends toward younger generations, more than a half (68.6%) fall in the 26–35 year-old range, and over three quarters (80.1%) in the 26–45 year-old range. Older people often do not purchase goods online and are less inclined to participate in surveys. The distribution of education tends to be highly educated, 52.6% of respondents are undergraduates, 29.5% of respondents are graduate or more. As a result that Wuhan is the capital city in the Hubei province, and most enterprises in Wuhan have higher academic requirements, 40.4% of respondents are medium income, with an earning of 762$ to 1523$ a month. High income (over 3046$ one month) and low income (below 457$) accounts for a small proportion. Moreover, the proportion of samples for shopping online for fresh food before the COVID-19 outbreak is 65.4%, never having purchased online fresh food is 34.6%.

### 3.2. Ordered Logit Regression Model

In the survey, respondents were asked to describe online and in-store shopping frequency, cost, and other associated information before the COVID-19 outbreak, during the COVID-19 pandemic, and after the COVID-19 pandemic. Therefore, the comparison between them will be made to analyze changes brought on by the COVID-19 pandemic. In addition, respondents were asked to express whether they would increase online shopping and to estimate the proportion of online shopping for fresh food after the COVID-19 pandemic. The options of proportion range from 1 (below 10%) to 6 (over 80%). We will use the ordered logit model for this analysis, which is widely used with variables that are ordered and multi-classified [52,53].

The ordered logit regression model is suitable for the case of ordered and multi-classified dependent variables, which can briefly be expressed as Equation (Equation 1) [54].
(1)$$p\left(Y_{i}>j\right)=\frac{exp(\alpha_{j}+\sum \beta_{j}\cdot x_{i})}{1+exp(\alpha_{j}+\sum \beta_{j}\cdot x_{i})}$$
where α is a constant and insignificant to factor $x_{i}$; $\beta_{i}$ is the regression coefficient of the independent variable; *X* is the set of independent variables; *Y* is the set of dependent variables; *p* is the the cumulative probability.

In the ordered logit model, the respondents’ online shopping proportion after COVID-19 pandemic was measured using a range from 1 (less than 10%, rarely buy fresh food online) to 6 (more than 80%, very frequent online shopping of fresh food), as the dependent variables. There are many factors influencing peoples’ online shopping proportion after the COVID-19 pandemic, such as personal attributes, shopping behavior characteristics before the COVID-19 outbreak, shopping behavior characteristics during the COVID-19 pandemic. These factors are independent variables. As for all independent variables, the items with a calibration value of 1 are set as the reference items. All the independent variables are selected with an introducing probability of 0.05 and rejecting probability of 0.1. R language will be used to perform ordered logit regression analysis on the data.

## 4. Results and Discussions

### 4.1. Characteristics of Shopping for Fresh Food

There are 54 respondents who did not buy fresh food online before the COVID-19 outbreak, which accounts for 34.6%. A total of 102 respondents (65.38%) have the experience of online shopping for fresh food. The majority do not exceed 30$ per online purchase, accounting for 90.2%. A total of 49.02% of respondents’ online shopping proportion was under 10%. Over half (63.72%) of them shop online for fresh food at least once a week. Therefore, the majority of them frequently shop online for fresh food, but the proportion of online shopping is very low. Nearly half (49.02%) of them have an online shopping proportion below 10% and the proportion of 10–20% accounts for 27.45%. Based on the data from Appendix A
Table A1, it can be inferred that most of the citizens bought fresh food in stores before the COVID-19 outbreak in Wuhan city. However, shopping frequency declined during the COVID-19 pandemic [14]. The most popular travel mode to stores for fresh food is walking, and car driving ranked the second. Nearly half of them go to stores for fresh food within 10 min. Most of them go to stores for fresh food within 20 min. The more shopping opportunities one can reach within 10 min, the less often one shops online [29]. This implies that Wuhan city has a high level of accessibility to grocery stores, and people will be more inclined to shop in-store. The results are consistent with the results of Bondemark [55].

However, during the COVID-19 pandemic, the indicators of characteristics are quite different, as citizens in Wuhan city stay at home and order fresh food online. They do not need to go out by car or other travel modes. Car driving declined, which accounts for 30.13% before the COVID-19 breakout. Online shopping could reduce personal trips [24,56]. Based on the data of Appendix A
Table A1, the majority of shopping online was for fresh food within 30 min (80.13%) and did not exceed 30$ (72.44%) per online purchase. Waiting was below 48 h (62.18%) and the fresh food will be delivered at home. What is more, respondents of 78.20% shopped online at least once a week. The proportion of over 80% for fresh food purchased online is 33.33%, while the proportion of below 10% is only 18.59%. During this hard time, people in other places donated fresh food to Wuhan city, the government organized volunteers to deliver it to citizens. If there was no donated free fresh food, the proportion of online shopping for fresh food would be higher. They prefer free choosing (68.59%) rather than food bags (containing several fresh foods and can not change). Even if the supermarket sells most of the fresh food in the form of the food bag, most people chose free selection rather than food bag.

### 4.2. Comparison of Shopping Behavior for Fresh Food before, during, and after the COVID-19 Pandemic

First, we compared online shopping frequencies before and during the COVID-19 pandemic. A total of 122 respondents shopped online for fresh food at least once a week during the COVID-19 pandemic, while 65 respondents did before the COVID-19 outbreak.

During the COVID-19 pandemic, citizens can not go out and must get food from online shopping or donation. From Table 3 and Table 4, associated with the detailed data of Appendix A
Table A1, we can compare the behaviors. Most of the respondents (78.20%) shopped online for fresh food at least once a week during the COVID-19 pandemic. Respondents of 9.86% spent over 30$ per order for fresh food purchased online before the COVID-19 outbreak, while the proportion is 27.56% during the COVID-19 pandemic. In order to reduce touch with each other and keep social distance, almost all fresh food shopping was done online. This contributes to the increase in the cost of online shopping. Despite the impact of the COVID-19 pandemic, there is an increase in logistics demand. People are shopping online more often than before. Even the speed of delivery was affected by the COVID-19 pandemic, the waiting time for fresh food after being ordered online within 12 h accounts for 27.56%, within 24 h is 62.18%, within 48 h is 91.67%. This means that most of them can get fresh food online within 2 days to guarantee the freshness of food in winter. Therefore, citizens in Wuhan may spend within 30$ and 30 min online shopping for fresh food once a week. The waiting time was about 24 h, which means they purchased fresh food that will be delivered at home the next day after being ordered online. This helps the citizens in Wuhan to get through the tough time. Even during the COVID-19 pandemic, the logistics system in Wuhan can operate efficiently.

The proportion of online shopping for fresh food increased during and after the COVID-19 pandemic, as shown in Table 5. Most of the respondents’ online shopping proportion was under 10% before the COVID-19 breakout, with 50 respondents. It means that the main way for fresh food is from in-store shopping before the COVID-19 breakout. During the COVID-19 pandemic, the option over 80% accounts for the largest proportion, with 52 respondents. More people shopped online during the COVID-19 pandemic, which is consistent with the results in Suceava County, Romania [7]. After the COVID-19 pandemic, more people tend to shop online for fresh food, and the number of people who chose the option over 10% has increased compared to before the COVID-19 breakout. Similar results are found by Eger et al., Mitchell et al., and Leone et al. [8,11,13].

The advantages and disadvantages of buying fresh food online from people’s perspective are shown in Appendix A
Table A1. No touching and saving travel time are the major advantages for fresh food online shopping as COVID-19 can be transmitted through contact [57]. Online shopping can reduce outdoor trips [24] and touch with each other, leading to more people buying online for fresh food. Some people think that online shopping is cheaper than in-store shopping [42], the after-sales service is timely and effective, these also promote people to buy online for fresh food often. Fresh food from online shopping is usually not as fresh as in-stores and is damaged during logistics transportation. This may be caused by packaging and slow logistics transportation. What is more, people can not see the real thing, and pictures can deceive people. These are the disadvantages of buying fresh food online, and the aspects that can improve the service of online fresh food shopping and encourage more people to buy fresh food online.

### 4.3. Model Results

The ordered logit model results of online shopping proportion for fresh food after the COVID-19 pandemic are shown in Table 6. As for independent variables, $P_{d}$ (proportion of online shopping for fresh food during the COVID-19 outbreak) is a significant variable for two types of people (people with or without the experience of online shopping for fresh food before the COVID-19 outbreak). For the proportion of fresh food bought online during the COVID-19 pandemic, the coefficients of people with and without online shopping experience for fresh food before the COVID-19 outbreak are 0.441 and 0.493, the odds ratio of them are exp(0.441) = 1.554 and exp(0.493) = 1.637. These imply that people’s proportion of buying online during the COVID-19 pandemic in Wuhan city increased by one unit. The odds of the higher proportion buying online after the COVID-19 pandemic for people with experience of online shopping for fresh food before the COVID-19 outbreak versus the people without experience of online shopping for fresh food before the COVID-19 outbreak are 1.554 and 1.637 times greater. Online shopping proportion before and during the COVID-19 pandemic has a positive effect on online shopping proportion after the COVID-19 pandemic. The results suggest that these people who have the experience of online shopping for fresh food before and during the COVID-19 pandemic will buy more online. Similarly, Zhai et al. argued that online experience has a positive effect on online shopping [58]. The COVID-19 pandemic has promoted more online shopping [7,8,13].

For people with experience online shopping for fresh food before the COVID-19 outbreak, age is transformed into two categories (over 45 as aged people, below 45 as young people). Age has a negative influence on online shopping, as shown in Table 6. Younger people are more inclined to shop online, so people (below 45 years old) are more likely to buy online, which is consistent with previous studies [7,14,29,30,38,56]. Online shopping frequency before the COVID-19 outbreak has a negative effect on online shopping proportion for fresh food after the COVID-19 pandemic, which means people with higher frequency buying online before the COVID-19 outbreak (also with higher online shopping proportion for fresh food before the COVID-19 outbreak) are more likely to increase online shopping proportion for fresh food after the COVID-19 pandemic. The result of low in-store food shopping frequency during the COVID-19 pandemic is consistent with previous studies [10]. The proportion of online shopping for fresh food before the COVID-19 outbreak has a positive effect on buying more online after the COVID-19 pandemic. The coefficient of $P_{b}$ (proportion of online shopping for fresh food before the COVID-19 outbreak) is 0.606, the odds ratio is exp(0.606) = 1.833. These imply that the likelihood of people’s online shopping proportion for fresh food before the COVID-19 outbreak increases by one unit, the odds of the higher online shopping proportion for fresh food after the COVID-19 pandemic versus the lower proportion is 1.833 times greater. People with a lower proportion of online shopping for fresh food before the COVID-19 outbreak will be more inclined to increase their online shopping proportion for fresh food after the COVID-19 pandemic.

As for people without experience of online shopping for fresh food before the COVID-19 outbreak, frequency of online shopping for fresh food during the COVID-19 pandemic has a negative effect on buying more online after the COVID-19 pandemic. This means people with higher frequency buying online during the COVID-19 pandemic (also with higher online shopping proportion for fresh food during the COVID-19 outbreak) will be more likely to increase online shopping proportion for fresh food after the COVID-19 pandemic. Travel time to the nearest store for fresh food has a positive effect on buying more online after the COVID-19 pandemic. The coefficient of $T_{b}$ (travel time to the nearest store for fresh food) is 0.580, the odds ratio is exp(0.580) = 1.786. These imply that the likelihood of people’s shopping time to stores for fresh food before the COVID-19 outbreak increases by one unit, the odds of the higher online shopping proportion for fresh food after the COVID-19 pandemic versus the lower proportion is 1.786 times greater. This means considering a long time trip to stores for fresh food, people will be more inclined to buy online.

## 5. Conclusions

There is a great increase in online shopping in China over these years. China’s online retail market is the largest in the world [22]. This paper aimed to explore the influences of the COVID-19 pandemic on fresh food shopping based on the survey data collected from Wuhan.

The COVID-19 pandemic has caused more people to buy fresh food online in Wuhan city, which is consistent with other countries [7,11,13,14]. The proportion of online purchases for fresh food is much higher than before and after the COVID-19 pandemic. The frequency and cost of online shopping during the COVID-19 pandemic are significantly higher than before the COVID-19 pandemic. The factors, such as the frequency of online shopping before the COVID-19 outbreak, frequency of online shopping during the COVID-19 pandemic, and age have a negative effect on the proportion of online shopping after the lock-down days, while the proportion of online shopping before the COVID-19 outbreak, the proportion of online shopping during the COVID-19 pandemic, and travel time of in-store shopping before the COVID-19 outbreak have a positive effect. People’s behavior changed by the influence of the COVID-19 pandemic. These has given online fresh food stores and logistics transportation companies an opportunity to provide better services to attract more customers.

City planners may set refrigerators in the communities, people can order online and get fresh food from refrigerators near their home. On the one hand, improving the shopping accessibility of fresh food to people, for example by providing refrigerators in the community, will motivate people to buy more often online and thus reduce travel trips, make a contribution to reducing air pollution. On the other hand, improving people’s online shopping satisfaction, for example through the improvement of packing of fresh food, will attract people to purchase online and hence elevate the economic prosperity of the cold chain industry. The government can give some policy support to encourage cold chain related companies to improve their service, making people’s lives more convenient.

Logistics companies could improve their operational capabilities, including reducing transportation time and improve packaging, ensuring the freshness of fresh food, and reducing the damage of goods during logistics transportation. Online retailers could provide items’ videos to customers, which would help reduce uncertainties. A virtual group so customers could communicate with each other should be provided as well, which would meet the social interaction needs. Online salesclerks should improve their communication skills to reduce customer’s uncertainties and enjoy shopping online. All these measures will ensure the freshness from online shopping as in-stores and narrow the gap between in-store and online shopping. As the threat of COVID-19 still exists, the logistics transportation companies should take disinfection measures to ensure food safety, which has been emphasized by previous studies [4,14]. Workers should be required to wear masks and gloves when working, the warehouse should be clean and tested regularly.

Future studies should address two limitations of this study. First, due to the impact of COVID-19, we conducted the survey online and received 156 effective questionnaires. Younger individuals are more likely to use the internet and participate in the survey. This introduce a bias that young people are more inclined to buy online. Future studies should add a face-to-face interview and get more samples after the COVID-19 pandemic. Second, as smart phones have become widely used, the Chinese are becoming more and more familiar with smartphone shopping, apps such as Jingdong, Taobao, Pinduoduo (e-retailer) are popular in China. Understanding how apps’ promotion affect people’s shopping decision would also be of interest, the future research can explore this interesting topic.

## Figures and Tables

**Figure 1 ijerph-18-04469-f001:**
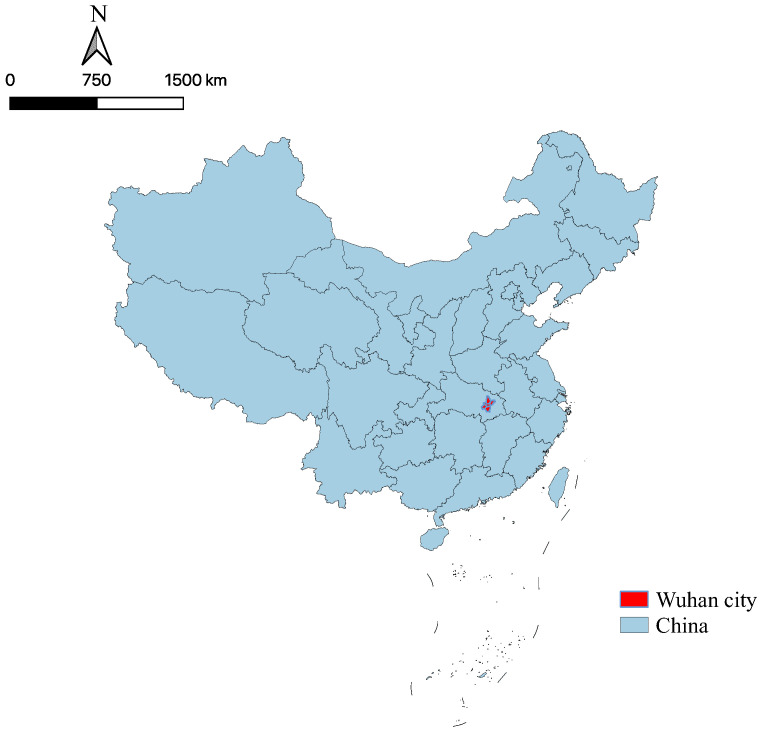
Location of Wuhan.

**Figure 2 ijerph-18-04469-f002:**
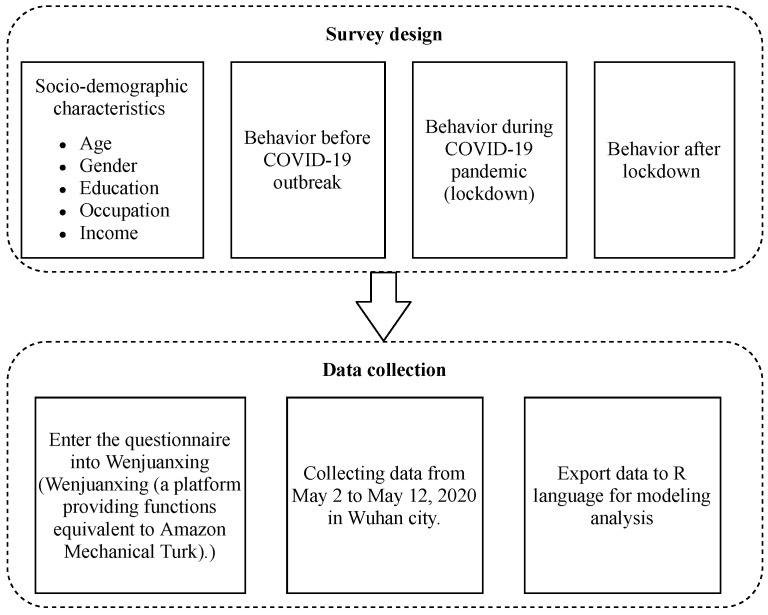
Procedures of survey design and data collection in the proposed research.

**Table 1 ijerph-18-04469-t001:** Comparison of in-store shopping and online shopping.

Attributes	Characteristics	
	In-Store Shopping	Online Shopping
Shopping	Travel cost [24,29]	No travel cost [24,29]
	Travel time [24,29]	No travel time [24,29]
	No need of internet [33]	Need internet [33]
	Shopping with social and leisure purpose [31,32]	No social purpose and less shopping fun [31,32]
Delivery	No delivery cost [34]	Delivery cost [34]
	No waiting time for delivery [34]	Waiting time for delivery [34]
Certainty	Touching items and more certainty [29]	Without touching and more uncertainty [29]

**Table 2 ijerph-18-04469-t002:** Summary of the variables of online shopping and in-store shopping literature review.

Study	Key Variables
[7,14,23,29,30,33,36,37,38,39,40]	Sociodemographic features (gender, education, income, car ownership, credit card ownership, household composition).
[7,14,24,25,29,35,37,40,41,42,42]	Shopping behaviors (frequency of online shopping and in-store shopping, shopping trip chaining, travel cost, travel time, waiting time, delivery cost, delivery way)
[29,33,35,36,37,39]	Internet behavior (frequency of Internet use, time on Internet, Internet connection type, number of years using Internet)
[29,37,40,41,43,44,45]	Shopping attitude (positive online or in-store shopping attitude, cost consciousness, quality consciousness, in-store shopping is fun, important to see products in person, lifestyle/personality, perception of prices, time pressure, adventure seeking)
[24,25,35,37]	Land use (shop access, urbanization level, living areas)

**Table 3 ijerph-18-04469-t003:** Frequency of online shopping for fresh food.

Proportion	Before COVID-19 Outbreak	During COVID-19 Pandemic
More than once a week	31	62
Once a week	34	60
Once every two weeks	10	19
Once a month	10	4
Less than once a month	17	11

**Table 4 ijerph-18-04469-t004:** Cost of online shopping for fresh food.

Cost ($)	<15	15–30	30–45	45–60	>60
Before COVID-19 outbreak	46	46	5	4	1
During COVID-19 pandemic	44	69	21	12	10

**Table 5 ijerph-18-04469-t005:** Online shopping proportion for fresh food.

Proportion	Before COVID-19 Outbreak	During COVID-19 Pandemic	After COVID-19 Pandemic
<10%	50	29	42
10–20%	28	20	46
20–40%	15	19	40
40–60%	6	21	17
60–80%	2	15	3
>80%	1	52	8

**Table 6 ijerph-18-04469-t006:** Ordered logit model of online shopping proportion for fresh food after lock-down days.

Attributes	Independent Variables	Online Shopping Proportion for Fresh Food after COVID-19 Pandemic		
		B	Std. Error	Signif.
People with experience online shopping for fresh food before the COVID-19 outbreak	Age	−1.144	0.582	0.046 *
	$F_{b}$	−0.421	0.157	0.006 **
	$P_{b}$	0.606	0.191	0.001 **
	$P_{d}$	0.441	0.112	4.623 × 10^−5^ ***
People without experience online shopping for fresh food before the COVID-19 outbreak	$F_{d}$	−0.444	0.270	0.092
	$T_{b}$	0.580	0.297	0.046 *
	$P_{d}$	0.493	0.161	0.001 **

*** *p*-value < 0.001, ** *p*-value < 0.01, * *p*-value < 0.05, *p*-value < 0.1. $F_{b}$: Online shopping frequency before the COVID-19 outbreak; $P_{b}$: Proportion of online shopping before the COVID-19 outbreak; $P_{d}$: Proportion of online shopping during the COVID-19 pandemic; $F_{d}$: Online shopping frequency before the COVID-19 pandemic; $T_{b}$: Travel time to the nearest store for fresh food before the COVID-19 outbreak (minutes).

## Data Availability

Not applicable.

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
