# Peer review of "Does COVID-19 Affect the Behavior of Buying Fresh Food? Evidence from Wuhan, China"

_ijerph, 2021, doi:10.3390/ijerph18094469_

Round 1
Reviewer 1 Report
The article is extremely relevant and well-written. It builds on a solid argumentation and develops in a fluent and clear way. It also highlights its limitations, and possible strategies to address them. Well done! I really enjoyed reading it.
Please note a glitch in the final part: "Future studies should address two limitations of this study", then there are actually three aspects listed. Thanks for editing it.
Author Response
The article is extremely relevant and well-written. It builds on a solid argumentation and develops in a fluent and clear way. It also highlights its limitations, and possible strategies to address them. Well done! I really enjoyed reading it.
Please note a glitch in the final part: "Future studies should address two limitations of this study", then there are actually three aspects listed. Thanks for editing it.
Response: Thank you for your professional suggestion. We have revised it. There are two limitations now. As shown below.
“Future studies should address two limitations of this study. First, due to the impact of COVID-19, we conducted the survey online and received 156 effective questionnaires. Future studies should add a face-to-face interview and get more samples after the COVID-19 pandemic. Second, as smartphones have become widely used and the Chinese are becoming more and more familiar with smartphone shopping, apps such as Jingdong, Taobao, Pinduoduo (e-retailer) are popular in China. Understanding how apps’ promotion affect people’s shopping decision would also be of interest, the future research can explore this interesting topic.”
Reviewer 2 Report
The authors of this study used a survey, descriptive analysis, and the ordered logit model to examine the behavior of buying fresh food online before, during, and after the COVID-19 pandemic. The study is good but still needs some improvements. Below I provide some comments to assess:
Major comments:
Title:
- “Does” not “Dose”.
Introduction:
- The introduction is good, the authors described the situation that leads people to buy more food online (specifically fresh food), and based on that they investigated in the paper the impact of the COVID-19 pandemic on buying fresh food online. However, in Lines (49-51), the authors stated “But there is no research on the impact of a public health issue on online shopping, and this paper aims to explore the impact of the COVID-19 pandemic on online shopping behavior for fresh food”. Not true, there is research deal with the effect of COVID-19 on online shopping behavior. Delete this phrase, and replace it with a small discussion of research investigating the changes in online shopping behavior during COVID-19, then, in the end, say “However, there is still a need for more research to investigate the effect of COVID-19 on buying fresh food online, based on that this paper aims ……”. In the following, I suggest a couple of papers you can use for this discussion.
- Leone, L. A., Fleischhacker, S., Anderson-Steeves, B., Harper, K., Winkler, M., Racine, E., ... & Gittelsohn, J. (2020). Healthy food retail during the COVID-19 pandemic: Challenges and future directions. International journal of environmental research and public health, 17(20), 7397.
- Alaimo, L. S., Fiore, M., & Galati, A. (2020). How the Covid-19 Pandemic Is Changing Online Food Shopping Human Behaviour in Italy. Sustainability, 12(22), 9594.
- Wang, E., An, N., Gao, Z., Kiprop, E., & Geng, X. (2020). Consumer food stockpiling behavior and willingness to pay for food reserves in COVID-19. Food Security, 12(4), 739-747.
Literature review:
- In “2.2. Online shopping and in-store shopping”, the authors have to make a more fair comparison between online and offline (in-store) shopping. Discuss better what you already listed in Table 1. Use previous literature to support the characteristics of each type of shopping, for example, for “Touching items and more certainty” you can use the following two papers. In fact, people buy fresh food online most probably because of the pandemic, otherwise, this type of product I believe people prefer to touch it and feel it. In the discussion section of your paper, you have also to get back and discuss this point and suggest recommendations to online retailers on how to maintain and improve their fresh food online business (Managerial implications - Suggestions on the policy).
- Elmashhara, M. G., & Soares, A. M. (2019). The impact of entertainment and social interaction with salespeople on mall shopper satisfaction. International Journal of Retail & Distribution Management, 47(2), 94-110.
- Koo, W., & Kim, Y. K. (2013). Impacts of store environmental cues on store love and loyalty: single-brand apparel retailers. Journal of International Consumer Marketing, 25(2), 94-106.
Methodology and discussion sections:
- No major comments.
Minor comments:
- The manuscript needs to undergo proofreading.
- Lines 31-32: The authors stated “People work from home, buy fresh food online, can not travel around during the lock days.”. Change “buy fresh food online” to “buy food online”. You are here still describing what happened during the lockdown in general, and in general, people bought everything online, all types of food not only fresh food.
- Line 127: The authors stated, “There are no effective antiviral drugs or vaccines against COVID-19”. I know you wrote this before the vaccination appeared. In this case update, then say something like “Even though we have vaccines now, still the numbers are raising ….”. Or you can keep the same statement but before that say “At the time of writing”.
- Lines 167-170: The authors stated, “Since there is no research directly investigate the effects of COVID-19 on fresh food shopping behavior, explore the effects of any virus or pandemic on shopping ways, this paper aims to fill this gap and provide evidence to support future research in fresh food shopping and cold chain logistics industry”. Don’t be very doubtless when you write such sentences. I searched on Google Scholars for three keywords “Fresh food + Online + COVID-19”. I got many results, many of them seem very relevant. So in this case, it is better to say something like “To the best of the authors’ knowledge, there is no research ….”. Or better to say something like “There is still limited research that investigates the effects of COVID-19 on fresh food shopping behavior” Then in the end discuss or refer to some studies that have already discussed the effect of COVID-19 on buying fresh food online. For example:
- Mitchell, R., Maull, R., Pearson, S., Brewer, S., & Collison, M. (2020). The impact of COVID-19 on the UK fresh food supply chain. arXiv preprint arXiv:2006.00279.
- Richards, T. J., & Rickard, B. (2020). COVID‐19 impact on fruit and vegetable markets. Canadian Journal of Agricultural Economics/Revue canadienne d'agroeconomie, 68(2), 189-194.
- Lines 241-243: You repeated the same idea twice in three lines.
- The explanation of Fb, Pb, Pd, Fd, Tb, Pd should be in the text, when you first mention them. I was so confused when reading them before I see this small note which comes after discussing them.
Good luck,
Author Response
Thank you for your professional suggestion. Followed your comments, I have learned a lot.
Reviewer 2’s Comments:
Title:
- “Does” not “Dose”.
Response: Thank you for your professional suggestion. We have revised it.
Introduction:
- The introduction is good, the authors described the situation that leads people to buy more food online (specifically fresh food), and based on that they investigated in the paper the impact of the COVID-19 pandemic on buying fresh food online. However, in Lines (49-51), the authors stated “But there is no research on the impact of a public health issue on online shopping, and this paper aims to explore the impact of the COVID-19 pandemic on online shopping behavior for fresh food”. Not true, there is research deal with the effect of COVID-19 on online shopping behavior. Delete this phrase, and replace it with a small discussion of research investigating the changes in online shopping behavior during COVID-19, then, in the end, say “However, there is still a need for more research to investigate the effect of COVID-19 on buying fresh food online, based on that this paper aims ……”. In the following, I suggest a couple of papers you can use for this discussion.
- Leone, L. A., Fleischhacker, S., Anderson-Steeves, B., Harper, K., Winkler, M., Racine, E., ... & Gittelsohn, J. (2020). Healthy food retail during the COVID-19 pandemic: Challenges and future directions. International journal of environmental research and public health, 17(20), 7397.
- Alaimo, L. S., Fiore, M., & Galati, A. (2020). How the Covid-19 Pandemic Is Changing Online Food Shopping Human Behaviour in Italy. Sustainability, 12(22), 9594.
- Wang, E., An, N., Gao, Z., Kiprop, E., & Geng, X. (2020). Consumer food stockpiling behavior and willingness to pay for food reserves in COVID-19. Food Security, 12(4), 739-747.
Response: Thank you for your professional suggestion. We have revised it and added these references. You can find it in lines 49-63. As shown below.
“As COVID-19 breakout all around the world, it changes people’s life, researchers show interest in it. People are more inclined to buy food online and the food reserve time extends from 3.37 to 7.37 days after COVID- 19 breakout [12]. Respondents’ education level, familiarity with buying food online Respondents’ education level, the experience of saving time buying food online, familiarity with buying food online, etc. affect the satisfaction of online shopping for food [13]. Leone et al. [14] show the challenges and propose future directions on the research of healthy food retail during the COVID-19 pandemic. They also emphasize that the effects of the COVID-19 pandemic on consumers’ shopping behavior still known little. Hence, there is still a need for more research to investigate the effect of COVID-19 on buying fresh food online. This paper aims to explore the effects of the COVID-19 pandemic on buying fresh food and how it works in Wuhan city. This paper will enrich the study of the effect of COVID-19 on buying fresh food, and the results will be useful for the cold chain logistics industry.”
Literature review:
- In “2.2. Online shopping and in-store shopping”, the authors have to make a more fair comparison between online and offline (in-store) shopping. Discuss better what you already listed in Table 1. Use previous literature to support the characteristics of each type of shopping, for example, for “Touching items and more certainty” you can use the following two papers. In fact, people buy fresh food online most probably because of the pandemic, otherwise, this type of product I believe people prefer to touch it and feel it. In the discussion section of your paper, you have also to get back and discuss this point and suggest recommendations to online retailers on how to maintain and improve their fresh food online business (Managerial implications - Suggestions on the policy).
- Elmashhara, M. G., & Soares, A. M. (2019). The impact of entertainment and social interaction with salespeople on mall shopper satisfaction. International Journal of Retail & Distribution Management, 47(2), 94-110.
- Koo, W., & Kim, Y. K. (2013). Impacts of store environmental cues on store love and loyalty: single-brand apparel retailers. Journal of International Consumer Marketing, 25(2), 94-106.
Response: Thank you for your professional suggestion. We have added more discussions and comparisons between online and offline (in-store) shopping. You can find it in section 2.2 (Online shopping and in-store shopping), lines 100-108. New suggestions give in section 4.4 (Suggestions on the policy), lines 391-400. As shown below.
“Entertainment and social interaction play an important role in customers' in-store shopping, which enhance customers' store loyalty as well [24,25]. Customers can feel and touch items and communicate with salesclerks, those who like shopping in-stores enjoy leisure time and social interaction there, while online shopping could not provide such experiences. However, there is no trips to stores by online shopping, travel time and cost saved [7]. So people can save time from shopping online to do other work. Online shopping could be a saving strategy and a leisure-oriented way.”
“Logistics company could improve their operational capabilities, including reducing transportation time and improve packaging, ensure the freshness of fresh food, and reduce the damage of goods during logistics transportation. Online retailers could provide items' videos to customers, which would help reduce uncertainties. A virtual group that customers could communicate with each other should be provided as well, which would meet the social interaction needs. Online salesclerks should improve their communication skills to reduce customer's uncertainties and enjoy shopping online. All these measures will ensure the freshness from online shopping as in-stores and narrow the gap between in-store and online shopping.”
Minor comments:
- The manuscript needs to undergo proofreading.
Response: Thank you for your professional suggestion. We have checked our article and revised it.
- Lines 31-32: The authors stated “People work from home, buy fresh food online, can not travel around during the lock days.”. Change “buy fresh food online” to “buy food online”. You are here still describing what happened during the lockdown in general, and in general, people bought everything online, all types of food not only fresh food.
Response: Thank you for your professional suggestion. We have revised it. As shown in lines 31-32. “People work from home, buy food online, can not travel around during the lock days.”
- Line 127: The authors stated, “There are no effective antiviral drugs or vaccines against COVID-19”. I know you wrote this before the vaccination appeared. In this case update, then say something like “Even though we have vaccines now, still the numbers are raising ….”. Or you can keep the same statement but before that say “At the time of writing”.
Response: Thank you for your professional suggestion. We have revised it. As shown in lines 146-148. “Even though we have vaccines now, still the numbers of infections and deaths are raising. More than 200 countries/regions have reported confirmed COVID- 19 cases, it turned out to be a major public health issue[2].”
- Lines 167-170: The authors stated, “Since there is no research directly investigate the effects of COVID-19 on fresh food shopping behavior, explore the effects of any virus or pandemic on shopping ways, this paper aims to fill this gap and provide evidence to support future research in fresh food shopping and cold chain logistics industry”. Don’t be very doubtless when you write such sentences. I searched on Google Scholars for three keywords “Fresh food + Online + COVID-19”. I got many results, many of them seem very relevant. So in this case, it is better to say something like “To the best of the authors’ knowledge, there is no research ….”. Or better to say something like “There is still limited research that investigates the effects of COVID-19 on fresh food shopping behavior” Then in the end discuss or refer to some studies that have already discussed the effect of COVID-19 on buying fresh food online. For example:
- Mitchell, R., Maull, R., Pearson, S., Brewer, S., & Collison, M. (2020). The impact of COVID-19 on the UK fresh food supply chain. arXiv preprint arXiv:2006.00279.
- Richards, T. J., & Rickard, B. (2020). COVID‐19 impact on fruit and vegetable markets. Canadian Journal of Agricultural Economics/Revue canadienne d'agroeconomie, 68(2), 189-194.
Response: Thank you for your professional suggestion. We have revised it. You can find it in lines 175-180. As shown below.
“The COVID-19 can be transmitted from food [46]. There is still limited research that investigates the effects of COVID-19 on fresh food shopping behavior. Mitchell, Maull, Pearson, Brewer, and Collison address that it is necessary to explore the impact of COVID-19 on the UK fresh food supply chain [47]. The Canadian fruit and vegetable markets affect by the COVID-19, and these changes may have a long-term effect on fresh fruit and vegetable distribution [48].”
- Lines 241-243: You repeated the same idea twice in three lines.
Response: Thank you for your professional suggestion. We have revised it. One of them is deleted. You can find it in lines 259-261. As shown below.
“There are 54 respondents who did not buy fresh food online before the COVID-19 outbreak, which accounts for 34.6%. 102 respondents (65.38%) have the experience of online shopping for fresh food. ”
- The explanation of Fb, Pb, Pd, Fd, Tb, Pd should be in the text, when you first mention them. I was so confused when reading them before I see this small note which comes after discussing them.
Response: Thank you for your professional suggestion. We have added the explanation when first mention them. You can find it in 4.3. Model results and Table 2.
“ Fb: Online shopping frequency before the COVID-19 outbreak; Pb: Proportion of online shopping before the COVID-19 outbreak; Pd: Proportion of online shopping during theCOVID-19 pandemic; Fd: Online shopping frequency before the COVID-19 pandemic; Tb: Travel time to the nearest store for fresh food before the COVID-19 outbreak (minutes).”
Reviewer 3 Report
(1) I think one of the many characteristics that good research should have is the possibility of reversal. In other words, the subject you want to study should be interesting and important to the literature and potential readers. However, if it is considered too natural or obvious without doing any research, it raises doubts about why this research should be done. This study is exactly like that. For those who are locked down due to COVID-19, they rely more on online for all their activities. This is true of all other activities, not just the purchase of fresh food. In addition, these symptoms are not limited to COVID-19, but are common to all people who come under quarantine. Therefore, the research questions that the authors are trying to clarify seem too natural and there is little additional value to be gained from the research.
(2) The research topic covers public health, online shopping, e-commerce, etc., but it is difficult to determine which academic field is suitable for the study because there is no focus on any of them.
(3) This study deals with the buying behavior of fresh food by people in quarantine. Therefore, this applies to a research in the field of management regardless of the department of the author's academic field, and accordingly, the social science empirical research method must be faithfully followed. Therefore, the research model should be presented more rigorous and detailed. In addition, conceptual and operational definitions for each concept should be presented more clearly. From this, a social science research model must be derived, and rigorous statistical analysis must be followed, not simply statistics at the level of frequency analysis. In that sense, the contribution to the literature is too minimal to be published in excellent journals such as IJERPH.
(4) The perspectives and focus of the authors on the reviewed previous studies are inconsistent and distracting. Therefore, it does not convincingly convey the value of comparison and research with previous studies. The authors should provide unpredictable insights and implications as to why buying fresh food by people quarantined under COVID-19 is worth researching. From that point of view, it would be better to add and reinforce previous studies.
(5) In the introduction, researchers should clearly clarify the gaps in previous studies that this study intends to fill and the knowledge to be added. It should also be clarified why this study is good and worth doing. Publishing cannot be justified simply because it has never been studied in the past. In addition, in the conclusion, the theoretical and practical contribution of this study should be summarized again. Based on this, theoretical and practical insights and implications for this study should be provided. Lastly, the limitations of this study and the direction of future research should also be clarified.
Author Response
(1) I think one of the many characteristics that good research should have is the possibility of reversal. In other words, the subject you want to study should be interesting and important to the literature and potential readers. However, if it is considered too natural or obvious without doing any research, it raises doubts about why this research should be done. This study is exactly like that. For those who are locked down due to COVID-19, they rely more on online for all their activities. This is true of all other activities, not just the purchase of fresh food. In addition, these symptoms are not limited to COVID-19, but are common to all people who come under quarantine. Therefore, the research questions that the authors are trying to clarify seem too natural and there is little additional value to be gained from the research.
Response: Thank you for your professional suggestion. We have explained why this research should be done in the Introduction. As shown below.
“As COVID-19 breakout all around the world, it changes people’s life, researchers show interest in it. People are more inclined to buy food online and the food reserve time extends from 3.37 to 7.37 days after COVID- 19 breakout [12]. Respondents’ education level, familiarity with buying food online Respondents’ education level, the experience of saving time buying food online, familiarity with buying food online, etc. affect the satisfaction of online shopping for food [13]. Leone et al. [14] show the challenges and propose future directions on the research of healthy food retail during the COVID-19 pandemic. They also emphasize that the effects of the COVID-19 pandemic on consumers’ shopping behavior still known little. Hence, there is still a need for more research to investigate the effect of COVID-19 on buying fresh food online. This paper aims to explore the effects of the COVID-19 pandemic on buying fresh food and how it works in Wuhan city. This paper will enrich the study of the effect of COVID-19 on buying fresh food, and the results will be useful for the cold chain logistics industry.”
(2) The research topic covers public health, online shopping, e-commerce, etc., but it is difficult to determine which academic field is suitable for the study because there is no focus on any of them.
Response: Thank you for your professional suggestion. The COVID-19 can be transmitted from food, it threatens people’s health. Based on this context, we try to explore the effects of the COVID-19 pandemic on fresh food shopping behavior.
(3) This study deals with the buying behavior of fresh food by people in quarantine. Therefore, this applies to a research in the field of management regardless of the department of the author's academic field, and accordingly, the social science empirical research method must be faithfully followed. Therefore, the research model should be presented more rigorous and detailed. In addition, conceptual and operational definitions for each concept should be presented more clearly. From this, a social science research model must be derived, and rigorous statistical analysis must be followed, not simply statistics at the level of frequency analysis. In that sense, the contribution to the literature is too minimal to be published in excellent journals such as IJERPH.
Response: Thank you for your professional suggestion. One of my academic fields is logistic behavior, which is consistent with this study. Further studies will consider to analysis the logistic behavior with a social science research model.
(4) The perspectives and focus of the authors on the reviewed previous studies are inconsistent and distracting. Therefore, it does not convincingly convey the value of comparison and research with previous studies. The authors should provide unpredictable insights and implications as to why buying fresh food by people quarantined under COVID-19 is worth researching. From that point of view, it would be better to add and reinforce previous studies.
Response: Thank you for your professional suggestion. We have added more previous studies. We have provided insights and implications in Introduction and Literature review. You can find it in these two sections. As shown below.
“Online shopping growing rapidly, its effects on in-store shopping have been studied by many researchers[6–8], as well as the relationship [8–11]. As COVID-19 breakout all around the world, it changes people’s life, researchers show interest in it. People are more inclined to buy food online and the food reserve time extends from 3.37 to 7.37 days after COVID- 19 breakout[12]. Respondents’ education level, familiarity with buying food online Respondents’ education level, the experience of saving time buying food online, familiarity with buying food online, etc. affect the satisfaction of online shopping for food[13]. Leone et al. [14]show the challenges and propose future directions on the research of healthy food retail during the COVID-19 pandemic. They also emphasize that the effects of the COVID-19 pandemic on consumers’ shopping behavior still known little.”
“The COVID-19 can be transmitted from food [46]. There is still limited research that investigates the effects of COVID-19 on fresh food shopping behavior. Mitchell, Maull, Pearson, Brewer, and Collison address that it is necessary to explore the impact of COVID-19 on the UK fresh food supply chain [47]. The Canadian fruit and vegetable markets affect by the COVID-19, and these changes may have a long-term effect on fresh fruit and vegetable distribution [48].”
(5) In the introduction, researchers should clearly clarify the gaps in previous studies that this study intends to fill and the knowledge to be added. It should also be clarified why this study is good and worth doing. Publishing cannot be justified simply because it has never been studied in the past. In addition, in the conclusion, the theoretical and practical contribution of this study should be summarized again. Based on this, theoretical and practical insights and implications for this study should be provided. Lastly, the limitations of this study and the direction of future research should also be clarified.
Response: Thank you for your professional suggestion. We clarify the gaps in previous studies and present this study intends in the Introduction. What’s more, we emphasize this in the last paragraph of the Literature Review. In the Conclusion, contributions, limitations of this study and the direction of future studies are detailed. As shown below.
“As COVID-19 breakout all around the world, it changes people’s life, researchers show interest in it. People are more inclined to buy food online and the food reserve time extends from 3.37 to 7.37 days after COVID- 19 breakout [12]. Respondents’ education level, familiarity with buying food online Respondents’ education level, the experience of saving time buying food online, familiarity with buying food online, etc. affect the satisfaction of online shopping for food [13]. Leone et al. [14] show the challenges and propose future directions on the research of healthy food retail during the COVID-19 pandemic. They also emphasize that the effects of the COVID-19 pandemic on consumers’ shopping behavior still known little. Hence, there is still a need for more research to investigate the effect of COVID-19 on buying fresh food online. This paper aims to explore the effects of the COVID-19 pandemic on buying fresh food and how it works in Wuhan city. This paper will enrich the study of the effect of COVID-19 on buying fresh food, and the results will be useful for the cold chain logistics industry.”
“Will the COVID-19 change behavior of buying fresh food? Will COVID-19 promote more often buying fresh food online? Is it possible for COVID-19 to change the way of shopping? This study aims to explore the effects of the COVID-19 pandemic on fresh food shopping behavior and try to answer the above questions. First, to describe the difference in fresh food shopping behavior before, during and after the COVID-19 pandemic. These included online and in-store shopping frequencies, costs and online shopping proportion, etc. Second, to explain how online shopping proportion after COVID-19 pandemic is influenced by shopping behaviors before and during COVID-19 pandemic.”
“After exploring and analyzing the influencing factors of people’s fresh food purchase behavior, suggestions are provided to improve online shopping services for fresh food. On the one hand, improving the shopping accessibility of fresh food to people, for example by providing refrigerators in the community, will motivate people to buy more often online and thus reduce travel trips, make a contribution to reducing air pollution. On the other hand, improving people's online shopping satisfaction, for example through the improvement of packing of fresh food, will attract people to purchase online and hence elevate the economic prosperity of the cold chain industry. The government can give some policy support to encourage cold chain related companies to improve their service, make people’s lives more convenient.”
“Future studies should address two limitations of this study. First, due to the impact of COVID-19, we conducted the survey online and received 156 effective questionnaires. Future studies should add a face-to-face interview and get more samples after the COVID-19 pandemic. Second, as smartphones have become widely used and the Chinese are becoming more and more familiar with smartphone shopping, apps such as Jingdong, Taobao, Pinduoduo (e-retailer) are popular in China. Understanding how apps’ promotion affect people’s shopping decision would also be of interest, the future research can explore this interesting topic.”
Reviewer 4 Report
I recommend the manuscript to be accepted after major revision done. The introduction should be redrafted to shorten, the literature review and data on the discussion of the results should be included in the appropriate chapter. The "Effects of Covid-19" section is unrelated to the research presented, nor is "Suggestions to the policy". Additionally, some minor errors were noticed:
1) page 2, lines 64-66: Information about the content of individual sections is unnecessary.
2) point 3.1.1: tabel 1 do not contain responders atributes;
3) The Authors give variable abbreviations of curriencies, it should be standarized and conversion to dollars should be given.
4) page 8, line 293: figures should be cited suquentially;
5) Figure 7: error in the inscription in the figure;
Author Response
Thank you for your professional suggestion. Followed your comments, I have learned a lot.

Reviewer 5 Report
Although the topic is interesting, the paper will need extensive revision before it can be considered for publication.
Abstract: What is meant by "experiences"?
The authors do not give any results - instead you have noted "will be discussed" and "suggestions are proposed" with no detail of the results and conclusions.
Introduction:
A lot of unnecessary detail is included about COVID - this is not the focus of your paper - there are countless papers highlighting the statistics (which have changed significantly since the paper was written). You should focus on the implications of COVID on shopping behaviour.
Similar statements made by different authors have not been integrated, resulting in repetition of facts and a lack of cohesion and logical flow.
Methods:
No detail on how the population and sample for the online surey were selected.
No detail on data collection procedures.
No detail on the design of the questionnaire - how are you sure your questions were valid and reliable?
No limitations are acknowledged - in my opinion, the fact that the data was collected via an online survey is a major limitation - those that use the internet to shop online will also be more likely to participate in the survey and this introduces a bias. The results are not representative of shoppers in Wuhan at all. The authors note that younger people tended to participate, confirming this bias.
Extremely small sample size for an online survey about shopping (n=171) - this complicates the applicability of the regression analysis.
Results:
Results depicted in both tables and multiple figures - very difficult to navigate and to make sense of the findings. For a publication it makes more sense to use tables to depict these type of findings.
Very limited discussion included - here you need to give reasons for findings and compare your results with other comparable and relevant publications.
Conclusion: A repetition of results that have already been reported.
Author Response

(The authors gave the same response as above.)

Round 2
Reviewer 2 Report
The authors
Author Response
Thank you for your professional suggestion.
Reviewer 3 Report
It seems that authors tried to revise the manuscript. However, for the reasons specified below, the extent and scope of the modifications are not considered sufficient. Therefore, the my decision is 'rejection' again.
(1) I don't think all papers should be published just because the authors studied the issues related to COVID-19. I agree that COVID-19 is the most pervasive and terrifying event in the world in the 21st century. However, I don't think that the pandemic of COVID-19 has historically been the first, and it will not be the last. Therefore, I do not believe that researching issues related to COVID-19 will increase the value of the research by itself. For whatever reason, when a person is in quarantine, it is no wonder that they become more dependent on online shopping for fresh food. Meanwhile, the authors say nothing about the difference between COVID-19-related isolation and isolation for other reasons. As such, I can never agree that this study adds valuable new knowledge to the literature. This is so easily inferred from past experiences in human history.
(2) Regardless of the author's major field or institution, it is very clear that this paper is a social science research. Therefore, in order to achieve the research goals, authors must follow a rigorous social science research methodology. However, it is considered that the research methodology of this study does not meet the rigor of the general level of research methodology of social science. While the authors' research may have some value, I don't think the level of this paper's quality could not justify the publication at the renowned journal IJERPH.
Author Response

(The authors gave the same response as above.)

Reviewer 4 Report
I recommend the article to be published. The applied corrections increased the value of the article.
Author Response
Thank you for your approval of our manuscript.
Reviewer 5 Report
Thank you for submitting a revised manuscript. The revised paper is improved, but the following issues remain relevant:
Extensive language editing is required
Introduction and literature review: Although certain sections have been removed, there is still a lack of integration and cohesion to clearly highlight the background, state the problem and highlight the value and implications of the current research.
Methods: Thank you for including a description of the questionnaire development based on variables that have been motivated in the literature. In my opinion, there is still not sufficient detail on how the population and sample for the online survey were selected, data collection procedures. Although limitations are now acknowledged, the fact that the
data was collected via an online survey is still a major limitation - the explanation given for a larger number of younger respondents has not been understood by the authors - I am very aware that younger individuals are more likely to use the internet (I was not questioning this fact), but it does introduce a bias in terms of how representative the sample is of all consumers who need to purchase fresh foods. Furthermore, the sample size remains extremely for an online survey about
shopping (n=171) - this continues to complicates the applicability of the regression analysis.
Results: Thank you for removing figures and replacing them with tables. I still feel that the discussion is limited in terms of implications of findings and comparison with other relevant studies.
Conclusion: This section is improved, but is still quite nonspecific.
Author Response
Thank you again for your positive comments and valuable suggestions to improve the quality of our manuscript.
